# Neural Edit Operations for Biological Sequences

**Satoshi Koide**
Toyota Central R&D Labs.
koide@mosk.tytlabs.co.jp

**Keisuke Kawano**
Toyota Central R&D Labs.
kawano@mosk.tytlabs.co.jp

**Takuro Kutsuna**
Toyota Central R&D Labs.
kutsuna@mosk.tytlabs.co.jp

## Abstract

The evolution of biological sequences, such as proteins or DNAs, is driven by the three basic *edit operations*: substitution, insertion, and deletion. Motivated by the recent progress of neural network models for biological tasks, we implement two neural network architectures that can treat such edit operations. The first proposal is the *edit invariant neural networks*, based on differentiable Needleman-Wunsch algorithms. The second is the use of deep CNNs with concatenations. Our analysis shows that CNNs can recognize *regular expressions without Kleene star*, and that deeper CNNs can recognize more complex regular expressions including the insertion/deletion of characters. The experimental results for the protein secondary structure prediction task suggest the importance of insertion/deletion. The test accuracy on the widely-used CB513 dataset is 71.5%, which is 1.2-points better than the current best result on non-ensemble models.

## 1  Introduction

Neural networks are now used in many applications, not limited to classical fields such as image processing, speech recognition, and natural language processing. Bioinformatics is becoming an important application field of neural networks. These biological applications are often implemented as a supervised learning model that takes a biological string (such as DNA or protein) as an input, and outputs the corresponding label(s), such as a protein secondary structure [13, 14, 15, 18, 19, 23, 24, 26], protein contact maps [4, 8], and genome accessibility [12].

*Invariance*, which forces a prediction model to satisfy a desirable property for a specific task, is important in neural networks. For example, CNNs with pooling layers capture the *shift invariant* property that is considered to be an important property for image recognition tasks. CNNs were first proposed to imitate the organization of the visual cortex [6]. This is often used to explain why CNNs work for visual tasks. Similarly, *rotation invariance* for image tasks is also studied [25]. Generally, it is important to model the proper invariances for a given application domain.

What is, then, the invariance in biological tasks? As is well known in bioinformatics, similar sequences tend to exhibit similar functions or structures (i.e., similar labels in terms of machine learning). Here, the similarity is evaluated by *sequence alignment*, which is closely related to the edit distance. This implies that labels associated with the biological sequences exhibit (weak) invariance with respect to a small number of edit operations, i.e., *substitution*, *insertion*, and *deletion*. This paper aims to incorporate such invariances, which we call *edit invariance*, into neural networks.

**Contribution.**   We consider two neural network architectures that incorporate the edit operations. First, we propose the *edit invariant neural networks*. This is obtained by interpreting the classical

Needleman-Wunsch algorithm [17] as a differentiable neural network. Next, we show that deep CNNs with concatenations can treat *regular expressions without Kleene stars*, indicating that such CNNs can capture edit operations including insertion/deletion. Our experiments demonstrate the validity of our approach. The test accuracy of protein secondary structure prediction on the widely-used CB513 dataset (e.g., [26]) results in 71.5% accuracy, the state-of-the-art performance compared to those of previous studies on non-ensemble models.

## 2   Edit Invariant Neural Networks (EINN)

**Differentiable Sequence Alignment.** In bioinformatics, sequence alignment is key in comparing two biological strings (e.g., DNA, proteins). The Needleman-Wunsch (NW) algorithm [17], a fundamental sequence alignment algorithm, calculates the similarity score between two strings on an alphabet $\Sigma$. As shown in Fig. 1, the similarity score is computed via dynamic programming to maximize the total score (illustrated as the double-lined square) by inserting or deleting characters (depicted as the vertical and horizontal arrows, respectively).

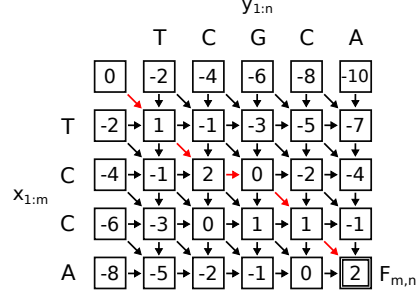

Figure 1: The NW alignment. The red line is a path that maximizes the score. The numbers in the cells correspond to $F_{i,j}$ in Algorithm 1.

Although the original NW algorithm is a function that uses strings as its arguments, we can naturally extend it to a differentiable function using embedding. Algorithm 1 shows the proposed dynamic programming procedure to calculate the NW score $s_{\mathrm{NW}}(x_{1:m}, y_{1:n}; g)$. Here, the scalar parameter $g$ is the gap cost that represents the cost to insert or delete a character. The differences from the original NW algorithm are three-fold.

1. The input sequences $x_{1:m} := [x_1, \cdots, x_m]$ and $y_{1:n} := [y_1, \cdots, y_n]$ are each $d$-dimensional time series (i.e., $x_i$ and $y_j$ are vectors in $\mathbb{R}^d$) of length $m$ and $n$, respectively.

2. Following the modification above, the score function is defined as the inner product instead of a predefined lookup table (Line 7).

3. The softmax function $\max^\gamma(x) = \gamma \log(\sum_i \exp(x_i/\gamma))$ is used instead of the hard max function (Line 10).

The dynamic programming in Algorithm 1 can be regarded as a computational graph, allowing us to differentiate the NW similarity score $s_{\mathrm{NW}}(x_{1:m}, y_{1:n}; g)$ with respect to $x_{1:m}$, $y_{1:n}$, and $g$. In principle, it is possible to apply automatic differentiation to obtain the gradient; however, automatic differentiation might be computationally expensive because exponentially there are many backward paths in the computational graph. To avoid this problem, we modify the backward computation by employing some algebraic substitutions. Consequently, we can compute the derivatives efficiently using dynamic programming, as shown in Algorithm 2 and Algorithm 3. See Appendix A in the supplementary material for the derivation of these algorithms. With the matrix $Q$ computed in Algorithm 2, we can calculate the derivative of the NW score with respect to $x_i$ and $y_j$ as follows.

---
**Algorithm 1:** Differentiable Needleman-Wunsch (forward): $s_{\mathrm{NW}}(x_{1:m}, y_{1:n}; g)$

---
1  $F \leftarrow 0$;     // (m+2)x(n+2) zero matrix
2  **for** $i = 0 \cdots m$ **do**
3  $\quad \lfloor F_{i,0} \leftarrow -ig$
4  **for** $j = 1 \cdots n$ **do**
5  $\quad F_{0,j} \leftarrow -jg$;
6  $\quad$ **for** $i = 1 \cdots m$ **do**
7  $\quad\quad a \leftarrow F_{i-1,j-1} + x_i \cdot y_j$;
8  $\quad\quad b \leftarrow F_{i-1,j} - g$;
9  $\quad\quad c \leftarrow F_{i,j-1} - g$;
10 $\quad\quad \lfloor F_{i,j} \leftarrow \max^\gamma(a, b, c)$
11 **return** $F_{m,n}$ **as** $s_{\mathrm{NW}}(x_{1:m}, y_{1:n}; g)$

---

$$\frac{\partial s_{\mathrm{NW}}}{\partial x_i} = \sum_{j=1}^{n} Q_{i,j} \exp(H_{i,j}/\gamma) \cdot y_j, \quad \frac{\partial s_{\mathrm{NW}}}{\partial y_j} = \sum_{i=1}^{m} Q_{i,j} \exp(H_{i,j}/\gamma) \cdot x_i, \qquad (1)$$

where $\quad H_{i,j} := F_{i-1,j-1} + x_i \cdot y_j - F_{i,j}$.

These derivatives are derived similarly to *SoftDTW* [3], a differentiable distance function corresponding to the dynamic time warping (hence the gapcost is not involved). For the derivative with respect to

| **Algorithm 2:** Calculation of $Q$ (backward). We denote $\varphi_\gamma(a,b) := \exp((a-b)/\gamma)$. | **Algorithm 3:** Calculation of $P$. We denote $\varphi_\gamma(a,b) := \exp((a-b)/\gamma)$. |
|---|---|
| 1   $Q \leftarrow 0$;    // (m+2) x (n+2) zero matrix <br> 2   **for** $i = 1 \cdots m$ **do** <br> 3     $\lfloor \; F_{i,n+1} \leftarrow \infty$ <br> 4   $F_{m+1,n+1} \leftarrow F_{m,n}$;    $Q_{m+1,n+1} \leftarrow 1$; <br> 5   **for** $j = n \cdots 1$ **do** <br> 6     $F_{m+1,j} \leftarrow \infty$; <br> 7     **for** $i = m \cdots 1$ **do** <br> 8       $a \leftarrow \varphi_\gamma(F_{i,j} + x_i \cdot y_j, F_{i+1,j+1})$; <br> 9       $b \leftarrow \varphi_\gamma(F_{i,j} - g, F_{i+1,j})$; <br> 10      $c \leftarrow \varphi_\gamma(F_{i,j} - g, F_{i,j+1})$; <br> 11      $Q_{i,j} \leftarrow aQ_{i+1,j+1} + bQ_{i+1,j} + cQ_{i,j+1}$ <br> 12   **return** $Q$ | 1   $P \leftarrow 0$;    // (m+2) x (n+2) zero matrix <br> 2   **for** $i = 0 \cdots m$ **do** <br> 3     $\lfloor \; P_{i,0} \leftarrow -i$ <br> 4   **for** $j = 1 \cdots n$ **do** <br> 5     $P_{0,j} \leftarrow -j$; <br> 6     **for** $i = 1 \cdots m$ **do** <br> 7       $a \leftarrow \varphi_\gamma(F_{i-1,j-1} + x_i \cdot y_j, F_{i,j})$; <br> 8       $b \leftarrow \varphi_\gamma(F_{i-1,j} - g, F_{i,j})$; <br> 9       $c \leftarrow \varphi_\gamma(F_{i,j-1} - g, F_{i,j})$; <br> 10      $P_{i,j} \leftarrow$ <br>         $aP_{i-1,j-1} + b(P_{i-1,j} - 1) + c(P_{i,j-1} - 1)$ <br> 11   **return** $P$ |

the gapcost $g$, we can derive it similarly using the matrix $P$ in Algorithm 3: $\frac{\partial s_{\mathrm{NW}}}{\partial g} = P_{m,n}$. As in the original NW algorithm, the proposed method can consider insertions/deletions. It is well known that the NW score is closely related to the edit distance. Given sequences $x_{1:m}$ and $y_{1:n}$, let us consider a modified sequence $x'_{1:(m-1)}$ where one feature vector $x_t$ is *deleted* from $x_{1:m}$. In such a case, the calculated scores $s_{\mathrm{NW}}(x, y)$ and $s_{\mathrm{NW}}(x', y)$ show a similar value. We call this property the *edit invariance*, which is expected to be important for tasks involving biological sequences.

**Convolutional EINN.** Here, we extend the traditional CNNs by the NW score introduced above. Let us consider an embedded sequence $X \in \mathbb{R}^{d \times L}$ of length $L$, and a convolutional filter $w \in \mathbb{R}^{d \times K}$ of kernel size $K$. Let $x \in \mathbb{R}^{d \times K}$ be a frame of length $K$ at a certain position in the embedded sequence $X$. In CNNs, the similarity is computed by the (Frobenius) inner product, i.e., $w \cdot x$. Our idea is to replace this inner-product-based similarity with the above proposed $s_{\mathrm{NW}}(x, w; g)$. Taking a limit as $g \to \infty$ (i.e., insertion and deletion are prohibited), $s_{\mathrm{NW}}$ is associated with a convolution as follows.

**Proposition 1.** For any $x \in \mathbb{R}^{d \times K}$ and any $w \in \mathbb{R}^{d \times K}$, we have $w \cdot x = \lim_{g \to \infty} s_{\mathrm{NW}}(x, w; g)$.

*Proof.* If $g \to \infty$, we have $F_{i,j} = F_{i-1,j-1} + w_i \cdot x_j$ in Line 10 of Algorithm 1. This leads to $\lim_{g \to \infty} s_{\mathrm{NW}}(x, w; g) = F_{K,K} = \sum_{i=1}^{K} w_i \cdot x_i = w \cdot x$.      $\square$

Therefore, the replacement of $w \cdot x$ in CNNs with $s_{\mathrm{NW}}(x, w; g)$ can be regarded as a generalization of CNNs. As mentioned above, the NW score is related to the edit distance, while the inner-product $w \cdot x$ corresponds to the Hamming distance, a special case of edit distance when insertion/deletion are prohibited (i.e., only substitutions are allowed). We also emphasize that this EINN-based convolutional architecture allows for the use of GPUs for batch, filter, and CNN-frame dimensions, although we cannot parallelize the innermost double loop of the dynamic programming.

**Summary.** In this section, we discussed a differentiable sequence alignment, EINN, to render the neural networks edit invariant. Subsequently, we proposed to replace the inner products in CNNs with EINNs. The proposed method is a generalization of CNNs, because the NW score $s_{\mathrm{NW}}$ converges to an inner product as $g \to \infty$ (Proposition 1). We employed a plain NW alignment in place of the inner product; however, there are many other alignment strategies, such as alignments with *affine gap costs*, *Smith-Waterman (SW) alignment* [22], and *BLAST* [1]. It is easy to replace the inner product in CNNs with an affine gap cost alignment or the SW alignment because these alignment methods are described as computational graphs with basic operations, such as '+' and 'max.' In contrast, creating a differentiable BLAST is highly challenging owing to its heuristic operations.

## 3   Deep CNN as a Regular Expression Recognizer

In bioinformatics, meaningful string patterns are called *motifs*, which resemble to regular expressions. For example, the N-glycosylation site motif is represented as N[^P][ST][^P], where N, P, S, and T

are amino acids, `[^P]` means any amino acid except for `P`, and `[ST]` means an amino acid `S` or `T`. This motif represents the following pattern: `N`, followed by any amino acid but `P`, followed by `S` or `T`, followed by anything but `P`. Another example is the C2H2-type zinc finger domain represented as `C-x(2,4)-C-x(3)-[LIVMFYWC]-x(8)-H-x(3,5)-H`, where `x(i)` and `x(j,k)` mean any sequence of length $i$ and any sequence of length between $j$ and $k$, respectively (we inserted "`-`" for readability).

We show the relationship between CNNs and regular expressions. First, we introduce *regular expressions without Kleene star*, which is a subset of the standard regular expressions.

**Definition 1.** The *regular expression without Kleene star* is a set of strings on an alphabet $\Sigma$ defined recursively as follows. First, the following are the regular expressions without Kleene star: 1) Empty set $\emptyset$; 2) Empty string $\varepsilon$; 3) A single character $\forall a \in \Sigma$. Next, let $R$ and $S$ be regular expressions without Kleene star. Then, the following sets of strings are also regular expressions without Kleene star. 4) *Concatenation* of strings in $R$ and $S$, denoted by $RS$. 5) A union of the sets $R$ and $S$, denoted by $R|S$ (called *alternation*). Moreover, given a string $q$, we say $q$ matches $R$ if $q$ is included in $R$.□

In short, this is equivalent to the standard regular expressions without the *Kleene star*, $R^*$, which accepts (potentially) infinite repeats of strings in $R$. Following this definition, we can easily confirm that the sets of strings represented by the two motifs mentioned above are the regular expressions without Kleene star. Hereafter, we use the Unix-like notations of regular expressions (see also the regular expression cheetsheet in the supplementary material (Appendix D)). For example, a regular expression `/a.b/` describes strings such as "`a`, followed by any character, followed by `b`." Furhter, `/a[bc]a/`, means strings such as "`a`, followed by `b` or `c`, followed by `a`," and `/(abc|ac)/` implies "`abc` or `ac`." It is noteworthy that the last regular expression `/(abc|ac)/` is equal to `/ab?c/`, where '`?`' means zero or one occurrence of the preceding token. Because the Kleene star "`*`" is not considered, regular expressions such as "`/ab*/`," describing "`a` followed by any number of `b`," are not considered.

**Simple regular expressions with CNNs.** Here, we reveal the relationship between regular expressions and CNNs. Let us start from a simple example to verify whether a given input string $x$ of length $L$ on an alphabet $\Sigma = \{$`a`, `b`, `c`$\}$ matches a regular expression `/abc/` for each position. We assume a one-hot representation for $x$, where each dimension corresponds to a character in $\Sigma$.

We compose a one-dimensional (1d)-convolutional layer whose filter matrix $w_1$ and bias $b_1$ are given by the one-hot representation $w_1 = (e_a, e_b, e_c)$ and $b_1 = -2$, respectively, where $e_a$ is the one-hot vector of character "`a`." This filter matrix, $w_1$, is shown in Fig. 2 (a). Using this filter, the output of the layer at position $i$ is 1 if $x_{(i-1):(i+1)}$ matches the regular expression `/abc/`, or smaller than 1 otherwise (see Fig. 2 (b)). Therefore, using ReLU (i.e., $relu(w_1 \cdot x_{(i-1):(i+1)} - b_1)$), we obtain 1 for matching and 0 for non-matching. This shows that we can emulate the exact pattern matching using a single 1d-convolutional layer. To simplify the discussion, we denote the convolution by a tuple $(w_1, b_1)$. Similarly, the recognizer for a regular expression `/ac/` can be emulated by a 1d-convolutional layer of kernel size $k = 3$ consisting of $w_2 = (e_a, e_c, 0)$ and $b_2 = -1$ (Fig. 2).

Next, let us use a regular expression `/ab?c/=/(abc|ac)/` as an example, which represents the pattern '`abc`', but accepts the *deletion* of the middle '`b`'. This can be recognized by the following multi-layer network. First, we apply the two convolutions above, $(w_1, b_1)$ and $(w_2, b_2)$. Then, using the outputs of these two filters as an input, the next convolutional layer of kernel size 1 with parameter $w_3 = (e_{abc} + e_{ac}) = [1, 1]^T$ and $b_3 = 0$ is applied (see the lowest matrix in Fig. 2 (b)).

**Relation between CNNs and regular expressions without Kleene star.** In principle, given a regular expression without Kleene star $R$, we can construct a two-layered convolutional network that accepts $R$ similarly. Let $k$ be the maximum length of strings in $R$. It is noteworthy that $k$ is finite because $R$ does not involve the Kleene stars. For the same reason, $R$ is a finite set. For each string $r$ in $R$, we construct a convolutional layer with kernel size $k$ accepting $r$. Subsequently, the outputs of these layers are input to the next convolutional layer of kernel size 1, which realizes the *OR* operation similarly to the filter $(w_3, b_3)$ above. This discussion leads to the following general proposition.

**Proposition 2** (CNN as a regular expression recognizer). Given a regular expression without Kleene star $R$, there exists a CNN that can verify whether a given string $x$ matches $R$ for each position of $x$.

Although this proposition demonstrates the potential ability of CNNs, the construction is inefficient especially when $|R|$ is large. For example, let us consider a regular expression `/ba./`, which consists

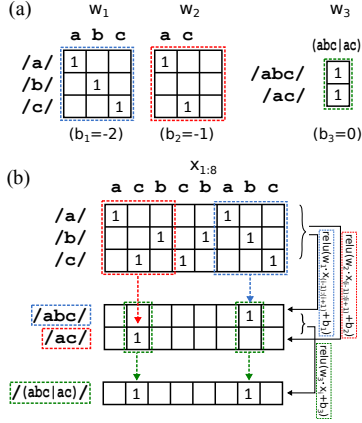

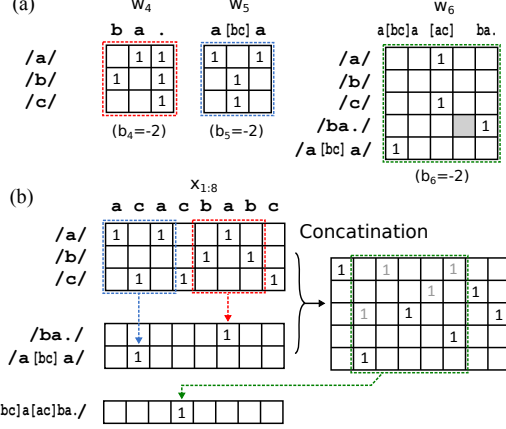

Figure 2: 1d-convolutional architecture accepting a regular expression /(abc|ac)/. (a) Weights of convolutions. (b) Applying to an example string `acbcbabc`. Note that blanks mean zero.

Figure 3: 1d-convolutional architecture accepting a regular expression /a[bc]a[ac]ba./. (a) Weights of convolutions. (b) Applying to an example string `acabaabc`. The 1's that do not match $w_6$ are grayed-out.

of $|\Sigma|$ strings. In this case, the construction above requires $|\Sigma|$ convolutional filters that might not be acceptable. In fact, it can be represented by only one filter, consisting of $w_4 = (e_b, e_a, e_a + e_b + e_c)$ and $b_4 = -2$ (Fig. 3 (a)). Furthermore, /a[bc]a/ corresponds to $w_5 = (e_a, e_b + e_c, e_a)$ and $b_5 = -2$ (Fig. 3 (a)). These examples show a possibility that a large regular expression $R$ could be compressed into a small CNN. We extend this discussion in the following.

**Going Deeper for Complex Regular Expressions.** According to Proposition 2, shallow yet wide neural networks can recognize an arbitrary regular expression without Kleene star $R$. Here, we discuss how the depth of the neural network relates to regular expressions. Further, we investigate the meaning of DenseNet [11] -like concatenation of the outputs from various layers.

Briefly, the depth and concatenation are important to obtain the *distributed representations* of string patterns, similar to that in image processing. Combining deep convolutions with concatenation, we can construct a recognizer for highly complicated regular expressions from small building blocks of simple regular expressions.

We explain based on an example using Fig. 3. In addition to the atomic regular expressions /a/, /b/, and /c/, let us consider two regular expressions, /a[bc]a/, and /ba./, as discussed above. Furthermore, we consider a regular expression /a[bc]a[ac]ba./, which is more complicated, yet is a combination of the simple regular expressions above. This regular expression can be divided into three parts: 1) /a[bc]a/ 2) /ba./ and 3) /[ac]/. The first two regular expressions can be recognized by $w_4$ and $w_5$ discussed above. To recognize the last one, /[ac]/, we employ the concatenation of two matrices, as shown in Fig. 3 (b). This allows us to combine with the atomic regular expressions. Concretely, using the convolutional filter $w_6$ shown in Fig. 3 (a) with $b = -2$, we can recognize /a[bc]a[ac]ba./ as shown in the lowest matrix in Fig. 3 (b). Similarly, if the shaded cell of $w_6$ in Fig. 3 (a) is set to 1, we can represent another regular expression with a *deletion* (i.e., /a[bc]a[ac]?ba./).

**Summary.** In this section, we first showed that even shallow CNNs can treat regular expressions without Kleene star in principle (Proposition 2). The number of filters can be, however, much larger than that in EINNs, which explicitly model edit operations. We provided an upper-bound of the filter size, $|R|$, although this bound is not very tight. This indicates that shallow CNNs cannot treat complex regular expressions efficiently. Then, we explained that the depth and concatenation can mitigate this issue. Concatenations allow us to reuse and combine simple regular expressions (like /a/, /b/, ...). In the next sections, we discuss and investigate what happens if concatenations are *not* used by comparing with the ResNet architecture, which does not involve concatenations. We also discuss the advantages and disadvantages of EINNs and CNNs.

## 4 Discussion

**CNNs and EINNs.**   In the previous sections, we have shown how to treat insertion and deletion of characters in a string, which is expected to be important for biological tasks. The EINNs treat insertions and deletions explicitly, while the deep CNNs with concatenation treat them implicitly. Here, we discuss their advantages and disadvantages. Although the EINNs model the well-established biological process directly through the NW algorithm, the computational cost is high due to the dynamic programming. Meanwhile, the computational cost of the deep CNNs is significantly lower. However, if the target regular expression involves many insertions and deletions, the number of convolutional filters required to represent it will increase rapidly. This is because the CNNs may have to treat such gapped patterns with separate convolutional filters, as shown in Fig. 2. One might wonder if we can mitigate this problem with pooling layers; however, we could not obtain improvements in the accuracy in a protein secondary structure prediction problem.

The CNN analysis in Section 3 was restricted to one-hot representations (called *binarized CNNs* herein). However, the filter weights and inputs of the *normal CNNs* are real numbers. We believe that this binarized analysis is still meaningful because such binarized CNNs are included in normal CNNs, indicating that the normal CNNs can learn more flexible patterns than the binarized CNNs.

**ResNet-like architecture.**   Finally, we discuss ResNet[10]-like architectures (i.e., using additive skip connections instead of dense connections). We argue that ResNet-like architectures are difficult to interpret. In fact, adding two matrices in the top and middle of Fig. 2 (b) generates matching results for /(a|abc)/ and /(b|ac)/ (here, we ignored the third row of the top matrix). This implies that additive skip connections do not allow us to combine simpler regular expressions freely. In our experiments, ResNet-like architectures do not demonstrate a better performance than DenseNet-like architectures for the protein secondary structure prediction task (see Section 6).

## 5 Related Work

**Sequence alignment and dynamic programming.**   The NW algorithm [17] is a fundamental sequence alignment algorithm. It is a global alignment algorithm, which aligns along the entire sequence. The Smith-Waterman (SW) algorithm [22], another well-known algorithm, is a local alignment algorithm where the subsequences are aligned.

Dynamic programming is used frequently for similarity computation between two sequences. *Dynamic time warping* (DTW) is often used for tasks involving time series (e.g., speech recognition [21]). Unlike the NW algorithm, DTW does not allow us to insert gaps (i.e., DTW does not consider the gap cost). Cuturi and Blondel [3] proposed a differentiable loss function called *Soft-DTW*. In speech recognition, *connectionist temporal classification* (CTC) is used as a loss function for two sequences [9]. The CTC explicitly models gaps differently from the NW algorithm. In bioinformatics, Saigo et al. [20] used a local alignment kernel, which is similar to SW alignment, to optimize amino acid substitution matrices by gradient descent. The gradient is computed similarly to EINNs, while embedding is not used. The difference between these methods and EINNs is that these methods are used as *loss functions*, whereas *EINNs are used as similarity functions in convolutions to make neural networks edit invariant*.

**NN as a language recognizer**   Thus far, the relationships between neural networks and the formal language theory have been studied in terms of RNNs. Minsky [16] demonstrated that any finite state machines can be emulated by a discrete state RNN with McCulloch-Pitts neurons. Forcada and Carrasco [5] considered a continuous version of the RNN, called the neural state machine. Gers and Schmidhuber [7] showed experimentally that the LSTM can learn context free grammar, including regular grammar. Unlike these studies, we focus on CNNs and reveal the relationship to regular expressions without Kleene star (Section 3).

**NN for biological sequences.**   Neural networks are used for several biological predictive tasks, such as protein secondary structure (shown below), protein contact maps [4, 8], and genome accessibility [12]. We herein focus on the protein secondary structure prediction problem, which is a sequence labeling problem predicting a label for each sequence position. The existing approaches are classified into three groups: 1) RNN-based models [13, 15, 23], 2) Hybrid of probabilistic models with neural

networks [18, 24, 26], and 3) CNN-based models [2, 14]. Among them, Li and Yu [13] reported the test accuracy of 69.4% for the CB513 dataset, a standard open dataset for this task, based on a bidirectional GRU model. Busia and Jaitly [2] reported the highest CB513 accuracy, 70.3% using a CNN-based model.[1] Based on the discussion in Section 3, we employ a much deeper architecture in our experiment. Consequently, we achieved a much higher accuracy, 71.5% (Section 6).

## 6   Experiments

In this section, we present the experimental results using a real task for biological sequences. We focus on the protein secondary structure prediction problem, which is widely studied both in the machine learning and bioinformatics communities. Overall, we will demonstrate that for protein structure prediction, it is important to adopt network architectures that consider insertions/deletions, as we have discussed previously.

**Dataset and implementation.**   We follow the previous studies for the secondary structure prediction. For the test, we used the widely-used CB513 dataset. For training, we used the filtered CB6133 dataset [13, 26], which has filtered out proteins in the original CB6133 dataset having 25% or higher similarity with some proteins in CB513. Consequently, the filtered CB6133 dataset includes 5534 proteins. We train the models that predict the eight-class secondary structure labels assigned at each position of a given sequence (i.e., a sequence labeling task). The feature vector at each position given in these datasets is the one-hot representation of amino acid (22-dim), and the position specific scoring matrix (PSSM, 22-dim). We employ zero-padding for convolutional operations to keep the sequence length constant. For implementations, we used PyTorch version 0.2. Unless otherwise noted, the default settings are used (e.g., weight initialization and hyperparameters for optimizers). The training was conducted on Nvidia Tesla GPUs.

**Results for simplified models.**   First, we investigate the effect of EINNs using simplified models and datasets. Here, two types of models are used: Tiny-CNN and Tiny-EINN. Figure 6 (a) shows the Tiny-CNN while the Tiny-EINN is obtained by replacing the Conv-5 layers with the EINN convolutional layer proposed in Section 2.

For training, we used the one-hot vector for input (i.e., PSSM is not used here), and 2% of training data (sequences) sampled from the filtered CB6133 dataset. We used the Adam optimizer with the minibatch size of 128, initial learning rate of 0.0002 (reduced by 1/10 at epoch 15), and weight decay $(10^{-5})$. We report the CB513 accuracy at epoch 30.

Table 1: Test accuracy (CB513).

| Method | Acc. (%) |
|---|---|
| Tiny-CNN | 42.0 |
| Tiny-EINN ($g = 2.5$) | 43.0 |

In Table 1, Tiny-EINN shows an accuracy that is 1.0-point better than Tiny-CNN. In this experiment, we used the fixed gapcost $g = 2.5$. Figure 4 shows how the accuracy changes with respect to $g$. We observe that, for $g > 10$, the accuracy is equal to that of the CNN, 42.0% (Proposition 1). Furhter, the maximum accuracy is achieved at $g = 2.5$, indicating the potential importance of insertion/deletion.

Next, we investigate what happens when different sizes of training data are used. Figure 5 shows the test accuracy for the CB513 dataset against the gapcost with the 1%, 2%, and 5% datasets. For the 5% dataset, the *performance gain*, defined by the difference between EINN (with $g$ at the peak) and CNNs (with $g \to \infty$), is 0.6%, which is smaller than that of the 2% dataset, i.e., 1.0%. For the 1% dataset, the performance gain is 1.4%, larger than that of the 2% dataset. To summarize, we obtained larger gains for smaller dataset. Thus, this result shows that the modeling power of EINNs is better than that of CNNs.

**Results for deeper models.**   Next, we show results for fully-deep models with realistic configurations, including a model achieving the state-of-the-art CB513 accuracy. Throughout the experiments, we used RMSProp for optimization, with the initial learning rate of 0.00033 and minibatch size of 8. The models are trained for 150 epochs, and the test accuracy at the last epoch is reported. We do not

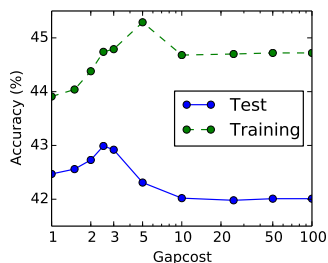
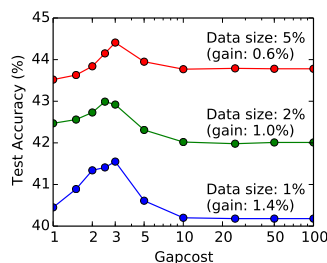

Figure 4: Gapcost $g$ vs accuracy (Tiny-EINN). Tiny-EINN is nearly equivalent to Tiny-CNN for $g > 10$.

Figure 5: Effect of data size. The performance gain of EINNs increases as the data size decreases.

use weight decay, and the learning rate is reduced by 1/10 at epoch 100. We do not employ other techniques including beam-search-based classification [2], or model ensembling [2, 13].

In addition, we found that data augmentation improves the accuracy. To create a new training data, we replaced the one-hot vector at randomly chosen positions with an amino acid drawn from the uniform distribution. In our experiments, we randomly replaced 15% of the residues. We maintained the PSSM dimension. This simple augmentation strategy improved the CB513 accuracy by up to 0.8-points. In the supplementary material (Appendix C), we investigate the effect of this augmentation. To the best of our knowledge, this technique has not been adopted in previous studies.

As a baseline, we stack the ConvBlocks shown in Fig. 6 (b). This is similar to the current state-of-the-art model proposed in [2]. Unlike their architecture, we do not employ nonlinearity after Conv-1 because we found it deteriorates the test performance when stacked deeply. We first apply two ConvBlocks. Then, at each position, a fully connected layer (of size 455) is applied, followed by batch normalization, dropout ($p = 0.2$) and ReLU. Finally, another fully-connected layer is applied to output the 8-class scores. To investigate the effect of the EINNs, we replace the convolutions shaded in Fig. 6 (b) in the *first* ConvBlock with EINNs of the same filter and kernel sizes.

The test accuracies for these models (2-block CNN[†] and 2-block EINN[†] in Table 2) indicate that the EINN-based model is again better, while the degree of improvement gets smaller. This can be interpreted as follows. Following the analysis in Section 3, the ConvBlock itself can recognize complex string patterns. This could reduce the need for EINNs, although it can potentially recognize complex patterns alone.

It is impossible to replace all of the convolutions in the model with EINNs owing to the following reasons. First, EINNs consume much more GPU memory than CNNs, thereby preventing us from applying them widely. Second, the computation time of the EINNs is slower than that of the CNNs. Although we have implemented EINNs using GPUs, as mentioned in Section 2, the computation speed is more than ten times slower than that of CNNs if the kernel size is $k = 5$. This is because the innermost double loop cannot be parallelized, thus resulting in a time complexity of $O(k^2)$, while the CNN computation runs in $O(1)$ time using GPUs. Hence, we investigate only CNNs in the following.

Table 2: Comparison of precisions for the secondary structure prediction on CB513 dataset. Note that these results are for non-ensemble models. (*: with multitasking / †: with data augment.)

| Method | Acc. (%) |
|---|---|
| Our 2-block CNN[†] | 69.7 |
| Our 2-block EINN[†] | 69.8 |
| Our 2-block CNN*[†] | 69.8 |
| Our 4-block CNN*[†] | 70.6 |
| Our 8-block CNN*[†] | 71.2 |
| Our 12-block CNN*[†] | **71.5** |
| Our 16-block CNN*[†] | 71.3 |
| Our 8-block MCNN*[†] | 71.3 |
| Our 12-block MCNN*[†] | **71.5** |
| ResNet*[†] (best result) | 71.0 |
| GSN [26] (2014) | 66.4 |
| DeepCNF [24] (2016) | 68.3 |
| DCRNN [13] (2016) | 69.4 |
| NextCond CNN [2] (2017) | 70.3 |

In Section 3, we argued that the 1) *depth* and 2) *concatenation* are important to handle edit operations with CNNs efficiently. In the following, we investigate the effect of each factor by

1) increasing the depth while keeping other conditions equivalent, and

2) using two different network architectures that do not involve concatenation (i.e., ResNet).

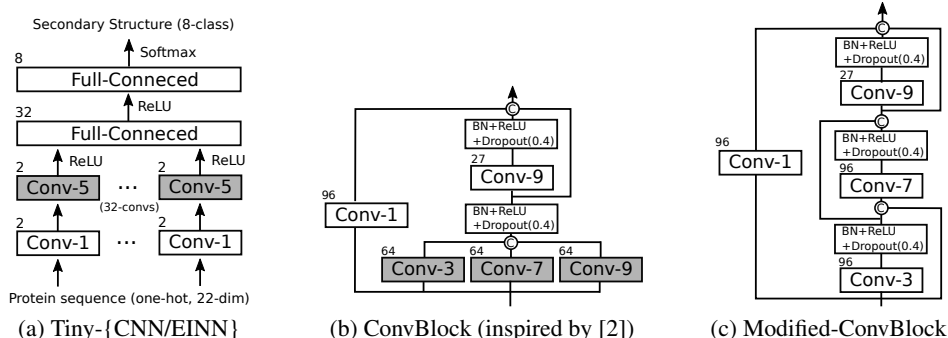

(a) Tiny-{CNN/EINN}   (b) ConvBlock (inspired by [2])   (c) Modified-ConvBlock

Figure 6: Network architectures. Conv-$k$ is the 1d-convolutional layer with kernel size $k$. The number at the top-left indicates the number of filters used. By replacing the shaded convolutions with EINN of the same kernel size, we obtain their EINN version. Here, Ⓒ means concatenation along the filter dimension. (a) 32-convs implies a grouped convolution with 32 groups. (b) This ConvBlock is stacked deeply. Then, at each position, a fully-connected layer is applied to output the 8-class scores (see the text for details.) (c) An alternative ConvBlock. This is to show the robustness against the network architecture.

**Effect of depth.** Next, we investigate the deeper CNN architectures. We begin with the shallow stacking of ConvBlocks, and make the stacking deeper (from 2 blocks to 16 blocks). The training procedure is the same as that in the previous experiment, except for the following points. First, we employed the widely-used multitasking technique [13, 19], simultaneously predicting the secondary structure (eight classes) and solvent accessibility (four classes). Second, we trained each model for 300 epochs, and the learning rate was reduced by 1/10 at 200 epochs.

As shown in Table 2, a 71.5% CB513 accuracy is achieved by our 12-block CNN*†, which is much higher than the results of the previous model, shown in the bottom of the table. In particular, it is more accurate than the previous best accuracy of 70.3%, for a single model [2]. Further, deeper models tend to show higher accuracy, which corresponds to the discussion in Section 3.

We can test other techniques such as ensemble models or templates [15] to improve the accuracy and avoid potential overfitting. Further, we should evaluate our model using various independent datasets and investigate other network architectures; however, we omitted most of them primarily because of resource limitations. In the following, we show what happens if different architectures are used.

**Effect of network architecture.** We investigate how network architecture affects performance by replacing the ConvBlock with the modified ConvBlock (Fig. 6 (c)), which also involves convolutional layers with concatenations. Note that the original and modified blocks have the same output dimensions and receptive fields. Table 2 ("8-block MCNN" and "12-block MCNN") shows that this modification does not change the accuracy. This indicates that there are many different architectures that can achieve the same performance, and there is still room for improvement.

Finally, we replace the ConvBlocks with the *residual blocks* [10], which we discussed in Section 4. Consequently, the best CB513 accuracy achieved by the ResNet-like models is 71.0% ('ResNet' in Table 2), and is worse than the models in Fig. 6 (b) and (c). For details, see Appendix B.

## 7   Conclusion

In this paper, we discussed how to make neural networks edit-invariant, a new important feature for ML tasks for biological sequences. First, we proposed EINNs that consisted of differentiable NW algorithm modules. Using EINNs as a generalization of CNNs, we confirmed that EINNs performed better than the corresponding CNN for a real biological task. This indicated that handling insertion/deletion was important for biological ML tasks. Next, we discussed that sufficiently deep CNNs with concatenation could emulate complex regular expressions. This implied that such deep CNNs could also treat the insertion/deletion of characters. Finally, for the protein secondary structure prediction task on the CB513 test dataset, the accuracy of our deep CNN was better than the current best result among the non-ensemble models.

## Footnotes

[1]This accuracy is based on a single model (i.e., non-ensemble model) prediction result. With model ensembling, they obtained 71.4%, which is comparable to our result, 71.5% (note that we do not use model ensembling in our experiment).

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
