[Supplementary Material]

# Supplementary material of "Neural Edit Operations for Biological Sequences"

## A  Derivatives of EINNs

Here, we compute the derivatives of the Needleman-Wunsch score, $s_{\text{NW}}(x_{1:m}, y_{1:n}; g)$. We use the notation in Algorithm 1. Since $s_{\text{NW}}(x_{1:m}, y_{1:n}; g) = F_{m,n}$, we consider $\partial F_{m,n}/\partial x_i$, $\partial F_{m,n}/\partial y_j$, and $\partial F_{m,n}/\partial g$.

Let us denote $Q_{i,j} = \partial F_{m,n}/\partial F_{i,j}$. We have

$$
\begin{aligned}
\frac{\partial F_{m,n}}{\partial F_{i,j}} &= \frac{\partial F_{m,n}}{\partial F_{i+1,j}} \frac{\partial F_{i+1,j}}{\partial F_{i,j}} + \frac{\partial F_{m,n}}{\partial F_{i,j+1}} \frac{\partial F_{i,j+1}}{\partial F_{i,j}} + \frac{\partial F_{m,n}}{\partial F_{i+1,j+1}} \frac{\partial F_{i+1,j+1}}{\partial F_{i,j}} \\
&= Q_{i+1,j} \cdot \frac{\partial F_{i+1,j}}{\partial F_{i,j}} + Q_{i,j+1} \cdot \frac{\partial F_{i,j+1}}{\partial F_{i,j}} + Q_{i+1,j+1} \cdot \frac{\partial F_{i+1,j+1}}{\partial F_{i,j}}.
\end{aligned}
\tag{2}
$$

In Algorithm 1, we have the following forward recursion:

$$
F_{i+1,j} = \max{}^\gamma(F_{i,j-1} + x_i \cdot y_j, F_{i,j} - g, F_{i+1,j-1} - g).
$$

The derivative of the softmax function $\max^\gamma(a, b, c)$ with respect to $b$ can be calculated as $\exp(\{b - \max^\gamma(a, b, c)\}/\gamma)$. Hence, we obtain

$$
\frac{\partial F_{i+1,j}}{\partial F_{i,j}} = \exp\left(\frac{F_{i,j} - g - F_{i+1,j}}{\gamma}\right) = \varphi_\gamma(F_{i,j} - g, F_{i+1,j}),
$$

which corresponds to $b$ in Line 10 of Algorithm 2. We can derive the remaining terms in Eq. (2) similarly.

Now, we compute $\partial s_{\text{NW}}/\partial x_i$, $\partial s_{\text{NW}}/\partial y_j$, and $\partial s_{\text{NW}}/\partial g$. Because of the chain rule, we have

$$
\begin{aligned}
\frac{\partial F_{m,n}}{\partial x_i} &= \sum_{j=1}^{n} \frac{\partial F_{m,n}}{\partial F_{i,j}} \frac{\partial F_{i,j}}{\partial x_i} \\
&= \sum_{j=1}^{n} Q_{i,j} \exp\left(\frac{F_{i-1,j-1} + x_i \cdot y_j - F_{i,j}}{\gamma}\right) \cdot y_j \\
&= \sum_{j=1}^{n} Q_{i,j} \exp(H_{i,j}/\gamma) \cdot y_j.
\end{aligned}
$$

We can derive $\partial s_{\text{NW}}/\partial y_j$ similarly. Finally, we consider $\partial s_{\text{NW}}/\partial g$. Following Algorithm 3, we define $a, b$ and $c$ as follows.

$$
a = \varphi_\gamma(F_{i-1,j-1} + x_i \cdot y_j, F_{i,j}), \quad b = \varphi_\gamma(F_{i-1,j} - g, F_{i,j}), \quad c = \varphi_\gamma(F_{i,j-1} - g, F_{i,j}).
$$

Based on the chain rule, we can calculate $\frac{\partial F_{i,j}}{\partial g}$ as follows.

$$
\begin{aligned}
P_{i,j} := \frac{\partial F_{i,j}}{\partial g} &= \frac{\partial}{\partial g} \max{}^\gamma(F_{i-1,j-1} + x_i \cdot y_j, F_{i-1,j} - g, F_{i,j-1} - g) \\
&= a \cdot \frac{\partial}{\partial g}\{F_{i-1,j-1} + x_i \cdot y_j\} + b \cdot \frac{\partial}{\partial g}\{F_{i-1,j} - g\} + c \cdot \frac{\partial}{\partial g}\{F_{i,j-1} - g\} \\
&= a \cdot P_{i-1,j-1} + b \cdot (P_{i-1,j} - 1) + c \cdot (P_{i,j-1} - 1)
\end{aligned}
$$

This corresponds to the recursive formula in Algorithm 3. Hence, we obtain $\partial s_{\text{NW}}/\partial g = \partial F_{m,n}/\partial g = P_{m,n}$.

# B    Results for ResNet-like architectures

Figure 7 shows the residual block (ResBlock) used in our experiments, which is similar to the ConvBlock (Fig. 6 (b)), but employs the additive skip connection. The number of filters $N$ is chosen from $\{48, 64, 96, 128\}$. In the skip connection of the first layer, we have to force the input/output dimensions the same. To this end, the input dimension size ($= 44$) is converted into $3N$ by applying Conv-1 in the skip connection. The training procedure used is equivalent to that used in our experiment for deep CNNs (i.e., 300-epoch training with RMSProp).

Figure 7: Residual blocks used in the experiment. $N$ is the number of filters.

Table 3 shows the result of our experiment. To control the model capacity, we tried different configurations by changing the number of blocks, the filter size $N$, and the weight decay parameter. Overall, the CB513 accuracies obtained in this experiment are by up to 71.0%, which are slightly lower than our best result obtained in Section 6 (71.5%).

Table 3: CB513 accuracy for ResNet-like models. Note that these results are for non-ensemble models. (*: with multitasking / †: with data augment / $N$: filter size / WD: Weight decay)

| Method | $N$ | WD | Acc. (%) |
|---|---|---|---|
| 4-ResBlocks*† | 128 | 0 | 70.8 |
| 8-ResBlocks*† | 128 | 0 | 70.8 |
| 12-ResBlocks*† | 128 | 0 | 70.4 |
| 16-ResBlocks*† | 128 | 0 | 70.4 |
| 8-ResBlocks*† | 96 | 0 | 70.7 |
| 8-ResBlocks*† | 64 | 0 | 70.6 |
| 8-ResBlocks*† | 48 | 0 | 70.7 |
| 4-ResBlocks*† | 128 | $10^{-6}$ | 70.6 |
| 4-ResBlocks*† | 128 | $10^{-5}$ | 70.6 |
| 4-ResBlocks*† | 128 | $10^{-4}$ | 70.5 |
| 4-ResBlocks*† | 128 | $10^{-3}$ | 70.5 |
| 8-ResBlocks*† | 128 | $10^{-6}$ | 70.8 |
| 8-ResBlocks*† | 128 | $10^{-5}$ | 70.9 |
| 8-ResBlocks*† | 128 | $10^{-4}$ | **71.0** |
| 8-ResBlocks*† | 128 | $10^{-3}$ | 70.9 |
| Our best result in Section 6 | – | 0 | (71.5) |

# C    Effect of data augmentation

To investigate the effect of the data augmentation mentioned in experiments, we show an evaluation result here. In this experiment, we employed our 8-block CNN architecture. When we changed the probability of random replacement, the results were as follows. Without noising (0%), we have 70.6% accuracy on CB513 dataset, and 71.0% (noise probability: 5%), 71.1% (10%), 71.2% (15%), 71.2% (20%), 71.1% (25%).

We consider that this improvement comes from "regularization effect." Random replacement forces the label-invariance around the training data. Accordingly, the predictor will be "flat", which could lead to good generalization.

## D  Regular Expression Cheetsheet

Table 3 shows the basic building blocks of regular expressions used in this paper.

Table 3: Regular expressions used in this paper.

| Regular Expression | Description |
| --- | --- |
| /a/ | Matches a single character, a |
| /abc/ | Matches a string, abc |
| /(abc\|ac)/ | Matches abc or ac |
| /./ | Matches a any single character |
| /[ab]/ | Matches a or b |
| /ab?/ | Matches a, followed by zero or one b |

Combining these regular expressions, we obtain more complex ones. For example, /a[bc]a[ac]?ba./ represents a set of strings that match

- a, , followed by b or c, followed by a, *optionally* followed by a or c, followed by ba, followed by any character.