[Reviews · NeurIPS 2018]

Reviewer 1



General conclusion. Good paper, interesting idea, slightly underwhelming results. I enjoyed reading it. The authors describe two neural architectures for sequence-to-sequence learning. And claim that this is specifically interesting for biological sequence tasks such as predicting secondary protein structure from the amino acids. The first is "Edit Invariant Neural Network" (EINN). The EINN is a generalization of a convolutional layer (1d convolutions over sequence steps) where filter elements can be "skipped" or can "skip" over input elements to maximize the matching between input and filter. The second is just a 1d convolutional network over sequence data, so not exactly new. The authors demonstrate that CNNs can act as matchers for star-free regular expressions, and that in some cases EINNs can outperform simple CNNs, although they can be much more computationally expensive. The paper seems to make two interesting contributions: - The description of an EINN layer to use for sequence matching, based on a novel differentiable variant of the Needleman-Wunsch algorithm. - The analysis of how CNNs can be used as learnable regular expression matchers The paper was generally well written, and the introduction of a differentiable sequence matching layer seems like it could be useful as the basis for future work. It is a bit unfortunate that the authors could not find a more scalable variant of EINN, since for the larger datasets EINN was not even used because it was too slow to run, so it just came down to running a plain cnn. It's also unfortunate that the improvements due to EINN are so minor - and leads one to wonder whether this idea is useful in practice. However, the novelty of introducing a new idea like this makes it, in my opinions, worth publishing, despite not having stunning results. It seemed like the EINN could have been slightly better described. Rather than include Algorithms 2 and 3, which are barely referenced in the text, and seem to be just the result of applying chain rule to Alg. 1 (unless i'm missing something), it would be good to have an algorithm which explicitely spells out the function of an einn layer. Note: Not sure if this is just a one-off problem, but in my printout many figure elements (e.g. the shaded box in Fig3, ) did not show up, which led to some confusion. Maybe check to make sure this is not happening to everyone. It might be helpful to have a small regular-expressions cheatsheet in the appendix. Minor notes by line: 21: cnns are not shift invariant. They are shift equivariant, and the pooling introduces some local shift invariance. 48: figure 1: should indicate in caption what red line represents, and that numbers in boxes are from alg. 1 72: Unless I'm not getting something, alg 2 and 3 just implement chain rule and would just be done internally by an automatic differentiation package, i dont understand the value of having them here. 72: You define phi_gamma(a,b) in the text but then only use them in alg 2,3, so maybe just put it in caption (if you keep algs 2,3 at all) 80: "show a similar value" should maybe be more specific - "do not differ by more than g"? Fig 3: explain greyed-out 1's 159,161: deepness -> depth 243: invariant -> constant 274: i don't know what residues are

Reviewer 2



Biological sequences which are close to each other in edit distance tend to have similar functional properties. The existing neural network-based solutions for biological applications have architectures that were originally proposed for image/speech processing applications and therefore do not leverage this property of biological sequences. The contributions of this paper are two-fold: (i) This paper proposes edit-invariant neural networks (EINNs), which have the ability to assign similar predictions to inputs that are close to each other in edit distance. (ii) This paper also shows some examples where deep CNNs with concatenations can handle edit operations well. The authors show numerical results for both these contributions on the protein secondary structure prediction problem. Strengths: • EINNs are a novel contribution. While these networks are a lot slower than CNNs due to the underlying dynamic programming operations, they can be expected to work better in biological applications where there is very little labeled data to work with (which can be the case often) due to their potentially superior modeling power. • The adaptation of the NW algorithm to make everything differentiable is done nicely. The argument that the proposed filter using the NW algorithm is a generalization of the standard convolutional filter is explained well. Weaknesses: • The numerical results shown in this work is extremely weak. o For the results shown in table 1, 2% of the available training data is used, and the accuracy improves from 42 to 43% when CNNs are replaced by EINNs. This is very unconvincing on two aspects: (i) The improvement is not sufficient enough to demonstrate the superiority of the EINNs. (ii) What happens when 1%, or 5%, or 10% of the training data is used? It would be better to show a curve here as opposed to just what happens at 2%. o The improvements shown in table 2 is very unconvincing as well. DenseNets were originally proposed to improve accuracies in image processing applications where edit distances are not used. An increase from 70.3% to 71.5%, using a more superior CNN with DenseNet-like concatenations, data augmentation, as well as multi-tasking by predicting solvent accessibilities is nice but hardly surprising. More precisely, I am not convinced that the improvement is due to better handling of edit operations which is the main purpose of this paper. • The section on deep CNNs with concatenations is motivated poorly. In particular, the following are unclear to me: o Is the paper arguing that CNNs can handle edit operations well? If yes, then what is the need for EINNs? o Is the paper arguing that CNNs with concatenations can handle edit operations better than CNNs without concatenations? If so, a more concrete analysis is necessary. o The few specific examples are informative, but one would need concrete bounds on the number of filters/ size of the models required for CNNs with or without concatenations to be able to make strong statements about CNNs handling edit operations. Minor comments: • Figure 3: “concatenation” is spelt incorrectly. • Figure 5: The legends need to be split into (a), (b) and (c) and explained in more detail. • Figure 5a: The architecture description is unclear. Specifically, what does the 32-convs represent? Isn’t the number on the top-left the number of filters?

Reviewer 3



This manuscript describes two methods for representing sequence editing operations in neural networks. The authors are motivated by problems in bioinformatics, in which insert, deletion and substitution operations represent basic moves in sequence evolution. The two techniques are complementary: one replaces the inner loop of the neural net training with a differentiable version of the Needleman-Wunsh dynamic programming algorithm; the other describes a general method for constructing convolutional network architectures that represent a particular class of regular expressions. Each approach has its drawbacks. The first method is elegant but computationally expensive; the second method naturally leads to very large networks, requiring some hoop-jumping in order to get the networks down to a more reasonable size. The authors demonstrate the utility of both methods on a small example, and then show that the CNN approach can lead to state-of-the-art results on the prediction of protein secondary structure. Overall, this manuscript was a pleasure to read, and the core ideas are compelling. It is certainly the case that deep architectures are being applied more and more widely within bioinformatics, and creative ways to represent sequence operations in this domain are likely to be of very wide interest. One concern I have is about the experimental results: the authors misrepresent their achievement relative to the state of the art. In the abstract they explicitly state that they have improved on the state of the art by 1.2% in accuracy. This is simply false, as pointed out in the footnote on p. 6: they have only improved on non-ensemble models by 1.2%. The best ensemble method achieves an accuracy only 0.1% less than the method reported here. The artificial choice to exclude ensemble methods is completely unmotivated. Indeed, there is no reason the authors could not have created ensembles of their own method. A second considerable caveat to the experimental results is that the authors have clearly and explicitly engaged in data snooping on the CB513 benchmark. The manuscript reports results from a wide variety of models on this benchmark, and presumably others were tried as well. As such, these results are likely overfit to this benchmark. To truly show that they have achieved the state of the art, the authors should have used an independent benchmark to do model development and then run only the best-performing method on CB513. One drawback to the first contribution (EINN) is that the Needleman-Wunsch algorithm is not, actually, widely used at all in bioinformatics. First, any DP algorithm for biosequence alignment needs to be generalized to use affine gap costs (i.e., charge more for introducing a new gap than for extending an existing gap). Second, only in rare cases is a global alignment required; it is much more common to use a local alignment algorithm like Smith-Waterman. The authors claim on line 211 that they can also apply the EINN method to SW, but this should be done or at least mentioned earlier. Third, in practice, both NW and SW are too slow for most bioinformatics applications, so heuristic approximations like BLAST are much more widely used. Given these caveats, the description of NW as "one of the most heavily-used sequence alignment algorithms" (line 43) is incorrect and misleading. Note that I don't see any conceptual reason why the proposed method could not be extended to affine gap penalties and local alignment; only that the authors should (1) admit that the method they have implemented falls short of capturing what is used in practice, and (2) at least sketch how the method could be extended to these other settings. Another concern regarding the empirical results is the "data augmentation" strategy proposed on lines 272-280. The method consists, essentially, of adding noisy versions of the training examples to the input. It is surprising that this method helps, and if this techniques actually accounts for up to 0.8% of the observed improvement, then it is just as important as the star-free regular expression CNN stuff that is purported to be the main advance in this part of the paper. The authors should discuss the motivation for this technique in more detail. The writing throughout is clear, though the text exhibits numerous grammatical and usage errors. I list some of these below, but the authors are encouraged to identify others as well. The definite or indefinite article is misused (included where not needed, skipped where needed, or wrong type of article) in lines 42, 63, 77, 107, 116, 126, 155, 231, 232. 12: delete "of" 79: Delete "of the." 137: "accepting" -> "accepts" 144: "star" -> "stars" 171, 203: The verb "allow" requires a direct object. 186: Semicolon before "however." 214: "neuron" -> "neurons" Figure 3: "Concatination" -> "Concatenation" 119: "does" -> "means" 242: "reminder" -> "remainder" 297: "of EINN" -> "for EINNs" 304: "indicates the time complexity" -> "yields a time complexity of" Other minor comments: 35: Briefly define "star-free regular expression" here. 37: State that CB513 is for protein secondary structure prediction. 101: Delete the sentence beginning "Therefore, ..." (It is obvious, given the previous sentence). 187 "Real biological tasks": Did the authors evaluate the method on tasks other than secondary structure prediction? Mention details here. I read the author response, and it seemed reasonable. I do think the paper has some problems with the empirical evaluation (as also commented on by Reviewer #2), but they are not fatal flaws. Hence, I did not modify my score.